# GENERALIZATION IN MONITORED MARKOV DECISION PROCESSES (MON-MDPS)

## ABSTRACT

Reinforcement learning (RL) typically models the interaction between the agent and environment as a Markov decision process (MDP), assuming fully observable rewards. In many real-world settings, this assumption fails, motivating the monitored Markov decision process (Mon-MDP), where rewards may be unobserved. Prior work on Mon-MDPs has been limited to simple, tabular cases, restricting their applicability to real-world problems. This work explores Mon-MDPs using function approximation and investigates the challenges. We show that combining function approximation with a learned reward model enables agents to generalize from monitored states with observable rewards to unmonitored states with unobservable rewards. Therefore, we demonstrate that such generalization with a reward model achieves near-optimal policies in environments formally defined as unsolvable. However, we also uncover a critical limitation: agents may incorrectly extrapolate rewards due to *overgeneralization*, which can lead to undesirable behaviors. To mitigate overgeneralization, we propose a cautious policy optimization method leveraging reward uncertainty. This work serves as a step towards bridging the gap between Mon-MDP theory and real-world applications.

## 1    INTRODUCTION

Reinforcement learning (RL) has emerged as a powerful framework for solving complex decision-making problems, achieving superhuman performance in a wide range of games, such as Atari games (Bellemare et al., 2013; Mnih et al., 2015), Go (Silver et al., 2016), Gran Turismo (Wurman et al., 2022), and poker (Moravčík et al., 2017). This success has fueled interest in applying RL to real-world applications, where RL has demonstrated remarkable capabilities. Notable examples include navigating stratospheric balloons (Bellemare et al., 2020), optimizing data center cooling (Evans et al., 2023), controlling water treatment (Janjua et al., 2024), and enabling robots to play table tennis semi-competitively (D'Ambrosio et al., 2024).

A common feature across these examples is that they modeled the interaction between the agent and environment as a Markov decision process (MDP), where the agent observes a state from the environment, takes an action, and receives an immediate reward as feedback. The agent aims to maximize the expected (discounted) cumulative reward. However, many real-world applications involve environments where rewards are not always observable. For example, consider an autonomous plant-watering robot responsible for maintaining plant hydration in a home. This robot learns through feedback, receiving positive reward from a human when it waters a plant, or using data from a monitoring system that measures soil moisture. However, continuous human feedback is often impractical due to time constraints, and installing sensors for each plant may not be cost-effective. Despite this, the agent should continue watering plants appropriately, even when rewards are unobservable, *e.g.*, when homeowners are away. To address the challenge of unobservable rewards, the monitored-MDPs (Mon-MDPs) framework was introduced (Parisi et al., 2024b), which models the interaction between the agent and environment when rewards are not always observable. However, prior work on Mon-MDPs (Parisi et al., 2024b;a; Kazemipour et al., 2025) has been limited to simple, tabular environments with a finite, tractable number of states and actions. This limitation highlights a gap between the current state of Mon-MDP research and its original motivation of modeling real-world applications where traditional MDPs cannot be applied.

This work takes an important first step towards bridging this gap by exploring Mon-MDPs in non-tabular settings, using function approximation (FA), and investigating the associated challenges with generalization. We demonstrate that while FA enables generalization, it can also lead to *overgeneralization*, where rewards are incorrectly extrapolated from monitored to unmonitored states, potentially leading to undesirable or unsafe behaviors. We provide a formal definition for *overgeneralization* and address it by leveraging reward uncertainty and robust policy optimization, showing that agents can *learn* to act cautiously in unmonitored states, mitigating unsafe behaviors.

This paper's key contributions are summarized as follows:

1. We demonstrate that incorporating FA and a learned reward model leads to near-optimal policies in some Mon-MDPs. In contrast, approaches that treat Mon-MDPs as traditional MDPs (*e.g.*, ignoring unobservable rewards or assuming they are zero) can yield sub-optimal policies even coupled with FA. This replicates observations of Parisi et al. (2024b) in tabular settings.

2. We empirically show that using FA with a reward model enables agents to generalize from monitored states with observable rewards to unmonitored states with unobservable rewards, allowing agents to achieve near-optimal performance in environments formally defined as unsolvable Mon-MDPs in the tabular setting.

3. We show that FA may lead agents to incorrectly extrapolate rewards, resulting in undesirable behaviors. We provide a formal definition of *overgeneralization* and adapt an algorithm that leverages reward uncertainty and robust policy optimization to address the dual challenges of reward observability and epistemic uncertainty. Our approach enables agents to *learn* to act cautiously when rewards are unobservable, mitigating overgeneralization.

This paper is structured as follows. Section 2 reviews background on MDPs, Monitored-MDPs, FA, and robust policy optimization. Section 3 introduces our approach, extending the reward model to non-tabular settings with FA. Experiments and results are presented in Section 4, demonstrating the effectiveness of our approach. Finally, Section 5 outlines our method's limitations and future work.

## 2 BACKGROUND

This section provides background on MDPs as a framework for modeling agent-environment interactions and introduces the Mon-MDPs framework, which extends MDPs to settings where rewards are not always observable. Additionally, we discuss the related work on generalization in RL using FA. Finally, we explore robust policy optimization techniques that address uncertainty challenges.

### 2.1 MARKOV DECISION PROCESSES

RL traditionally frames the interaction between the agent and environment as an MDP. An MDP is a mathematical framework for sequential decision-making, defined by a tuple $(\mathcal{S}, \mathcal{A}, \mathcal{R}, \mathcal{P}, \gamma)$. At each step $t$, the agent receives a state from the environment $s_t \in \mathcal{S}$, takes an action $a_t \in \mathcal{A}$, and receives a single scalar reward $r_t \in \mathbb{R}$ sampled from the reward function $r_t \sim \mathcal{R}(s_t, a_t)$. The agent then transitions to the next state $s_{t+1} \in \mathcal{S}$ according to a Markovian state transition probability distribution $s_{t+1} \sim \mathcal{P}(s_t, a_t)$. Therefore, the agent acts in the environment according to a policy, which probabilistically maps a state to an action $\pi(a_t|s_t)$, as shown in Figure 1. To evaluate and rank policies, a state-action value (Q-value) is defined as the expected discounted return of being in state $s$ and taking action $a$ under policy $\pi$, denoted as, $Q^\pi(s,a) = \mathbb{E}_\pi \left[ \sum_{t=1}^\infty \gamma^{t-1} r_t | s_t = s, a_t = a \right]$. According to the Bellman optimality equation (Bellman & Kalaba, 1957), there is a unique optimal value function for each MDP $Q_*^\pi(s,a) = \sum_{s_{t+1}} \mathcal{P}(s_{t+1}|s_t, a_t) \left[ \mathcal{R}(r_t|s_t, a_t) + \gamma \max_{a_{t+1}} Q_*^\pi(s_{t+1}, a_{t+1}) \right]$, and, there is at least one optimal policy $\pi_*(s_t) = arg\,max_{a_t} Q_t(s_t, a_t), \forall s_t \in \mathcal{S}$.

### 2.2 MONITORED MARKOV DECISION PROCESSES

Many real-world applications cannot be modeled as traditional MDPs because rewards are not always observable by the agent. The Mon-MDP framework (Parisi et al., 2024b) addresses this limitation by defining the agent-environment interaction as involving both an environment MDP and

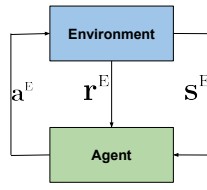 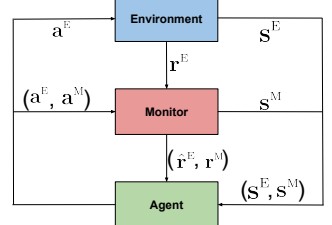

Figure 1: (Left) MDP diagram, (Right) Monitored-MDP diagram.

a separate monitor MDP that controls when the reward is observed (see Figure 1). Consider the tuple $(\mathcal{S}^E, \mathcal{A}^E, \mathcal{P}^E, \mathcal{R}^E, \gamma, \mathcal{M}, \mathcal{S}^M, \mathcal{A}^M, \mathcal{P}^M, \mathcal{R}^M)$, where $(\mathcal{S}^E, \mathcal{A}^E, \mathcal{P}^E, \mathcal{R}^E, \gamma)$ represents the environment MDP, whereas $(\mathcal{S}^M, \mathcal{A}^M, \mathcal{P}^M, \mathcal{R}^M)$ represents the monitor MDP[1].

The transition in Mon-MDPs can be divided into a transition to the next environment state, depending on the current environment state and action $s_{t+1}^E \sim \mathcal{P}^E(s_t^E, a_t^E)$, and a transition to the next monitor state, which depends on both the environment and monitor state-action $s_{t+1}^M \sim \mathcal{P}^M(s_t^E, s_t^M, a_t^E, a_t^M)$. Unlike traditional MDPs, the agent does not directly observe the reward from the environment, but instead observes the *proxy reward*, which is generated by a Markovian monitor function $\mathcal{M}$. The proxy reward denoted as $\hat{r}_t^E \sim \mathcal{M}(r_t^E, s_t^M, a_t^M)$, can be either observable $\hat{r}_t^E \in \mathbb{R}$ or unobservable $\hat{r}_t^E = \perp$. Therefore, at any timestep $t$, the agent receives the environment and monitor states $(s_t^E, s_t^M)$, takes actions in both the environment and monitor $(a_t^E, a_t^M)$, and receives a tuple of a proxy reward and a monitoring reward $(\hat{r}_t^E, r_t^M)$. However, the agent's goal is to maximize $\mathbb{E}_\pi \left[ \sum_{t=1}^\infty \gamma^{t-1}(r_t^E + r_t^M) \right]$, i.e., the discounted sum of monitoring reward and the actual (possibly unobserved) environment reward[2]. In this sense, MDPs are considered a special case of Mon-MDPs, where the reward is always observable $\hat{r}^E = r^E$, the monitoring reward $r^M = 0$, and there is a single monitoring state and action $|\mathcal{A}^M| = |\mathcal{S}^M| = 1$.

Mon-MDPs can be categorized into two distinct types based on the existence of a discernible optimal policy (Parisi et al., 2024b): i) *solvable Mon-MDPs*, where at least one optimal policy exists for all indistinguishable reward functions; ii) *unsolvable Mon-MDPs*, where there is no policy that is optimal for all indistinguishable reward functions (*i.e.*, the agent can never know if its policy is optimal). An extreme case of unsolvable Mon-MDPs is *hopeless Mon-MDPs*, where all policies are optimal for some indistinguishable reward function (*i.e.*, any policy for the agent could be optimal). An example of a hopeless Mon-MDP is a scenario where rewards are never observed.

To address the challenge of reward observability in Mon-MDPs, Parisi et al. (2024b) proposed learning a tabular reward model that estimates the environment reward and incorporates the estimated values into the value function update. While this approach is theoretically proven to converge to an optimal policy for solvable Mon-MDPs in tabular settings, the tabular reward model requires states and actions to be finite, with the reward model and value functions explicitly represented as tables. Specifically, the tabular reward model convergence guarantees hold only if: i) the environment is tabular with a finite or countably infinite (Halmos, 1960) number of states and actions; ii) each joint state pair $(s^E, s^M)$ can be visited infinitely often given infinite exploration; iii) the agent can observe the environment reward for each environment state-action pair with some nonzero probability; and iv) the proxy reward is truthful, meaning $\hat{r}_t^E = r^E$ or $\hat{r}_t^E = \perp \; \forall t$. These assumptions reveal significant limitations of the tabular reward model: many real-world applications involve non-tabular or infinite state spaces, where ensuring sufficient visitation of all state-action pairs becomes infeasible, and further complicates the exploration-exploitation trade-off. Nonetheless, requiring the agent to observe the environment reward for every state-action pair further constrains the reward model's applicability. To overcome these limitations and relax these restrictive assumptions, the reward model must be extended using FA, enabling learning in complex, non-tabular environments.

## 2.3 FUNCTION APPROXIMATION (FA)

Many real-world problems involve high-dimensional, possibly continuous states or actions. To address this complexity, FA aims to automatically learn to identify similarities, differences, and rela-

---

[1]Mon-MDPs may appear similar to POMDPs (Åström, 1965), the key distinction is that in POMDPs, states may not be fully observed but rewards are, whereas in Mon-MDPs, rewards themselves may not be observed.

[2]Table 2 in the appendix provides a summary table of the notation used throughout the paper.

tionships between states and actions. A function approximator is denoted as $y = f(x, \theta)$, where $y$ is the target, $x$ is the input features, and $\theta$ are the function's parameters. The ultimate goal of FA is not simply to memorize examples from the training data but to capture the underlying data-generating process, thereby enabling generalization to unseen examples.

FA helps RL handle high-dimensional states (*e.g.*, images) and large or continuous action spaces, enabling generalization and scaling to complex environments where tabular methods are impractical. FA can be applied in several ways. First, approximate the action-value function, where $\hat{Q}(s, a, \theta) \approx Q(s, a)$ and $\theta \in \mathbb{R}^d$, with $d \ll |\mathcal{S}|$. Such an approximation can be achieved using either linear methods (Sutton, 1988; Boyan & Moore, 1994; Gordon, 1995) or non-linear methods (Ernst et al., 2005; Lange et al., 2012; Mnih et al., 2013). Second, parameterize the policy, presenting the policy as a distribution $a \sim \pi(s, \theta)$ (Konda & Tsitsiklis, 1999; Sutton et al., 2000; Schulman et al., 2015; 2017; Haarnoja et al., 2018). Third, learn environment models, including the reward function $\hat{\mathcal{R}}(s, a, \theta)$ and transition function $\hat{\mathcal{P}}(s, a, \theta)$ (Kuvayev & Sutton, 1996; Sutton et al., 2008; Deisenroth & Rasmussen, 2011; Silver et al., 2016; Hafner et al., 2020).

### 2.4 ROBUST POLICY OPTIMIZATION

Despite FA's ability to handle high-dimensional data and generalize, it may produce incorrect predictions, particularly for novel, out-of-distribution examples. Such errors are a result of *epistemic uncertainty*, which arises from insufficient knowledge due to a lack of data from such examples. This is different from *aleatoric uncertainty*, where some examples lead to stochastic outcomes.

In RL, various approaches address uncertainty through robust policy optimization, enabling agents to learn policies that perform well under uncertainty, adversarial conditions, or distributional shifts (Russel et al., 2020; Jiang et al., 2021). To specifically address epistemic uncertainty, the "learning to be cautious" framework was initially proposed for a risk-averse approach within MDPs (Mohammedalamen et al., 2021). This framework combines an ensemble of neural networks to quantify epistemic uncertainty (Tibshirani, 1996; Heskes, 1996; Lu & Van Roy, 2017; Pearce et al., 2018; Osband et al., 2019; Lee et al., 2021) and applies conditional value at risk (CVaR) optimization, utilizing the $k$-of-$N$ counterfactual regret minimization (CFR) (Chen & Bowling, 2012). This method enables agents to learn to act cautiously in uncertain environments, where $k$ and $N$ are integers, $k > 0$, $N \geq k$, and the $k/n$ ratio controls the robustness level: $k = N$ corresponds to risk-neutral behavior, while a smaller $k$ with a larger $N$ induces a more risk-averse policy. Specifically, at each iteration, the $k$-of-$N$ CFR algorithm samples $N$ reward models and updates the policy based on the average performance of the $k$-worst models.

Although the "learning to be cautious" framework was originally designed for traditional MDPs, we extend it to Mon-MDPs to introduce risk aversion for novel states, where we may never obtain an accurate reward estimate.

## 3 REWARD MODEL WITH FUNCTION APPROXIMATION

To enable agents to generalize in Mon-MDPs, we extend the previously proposed tabular reward model (Parisi et al., 2024b) by incorporating FA. This extended reward model is denoted as $\hat{\mathcal{R}}(s^E, a^E, \theta)$, which takes the environment state and action $(s^E, a^E)$ as input, and predicts the environment reward. The reward model is trained using samples where the proxy reward equals the environment reward $\hat{r}^E = r^E$. In addition to the reward model, we also train a Q-model, denoted as $\hat{Q}_\pi(s^E, s^M, a^E, a^M, \theta)$, incorporating both the environment and monitor states as input $(s^E, s^M)$ to estimate the Q-value for each joint environment-monitor action $(a^E, a^M)$. Both the reward model and Q-model are implemented using a neural network architecture composed of two convolutional layers followed by two fully-connected layers. The Q-model follows the deep Q-Network (DQN) approach (Mnih et al., 2013), which employs a target network and a Q-network, trained using mini-batches from a replay experience buffer. For exploration, we use $\epsilon$-greedy with $\epsilon$ linearly decayed[3]. Further details on the experimental setup, model training procedure, computational resources, and hyper-parameters tuning are provided in the Appendix A. Within this framework, our implementation extends the original DQN in two key ways: i) DQN receives the joint state $(s^E, s^M)$ as input

---

[3]Our code is available anonymously with the supplementary materials.

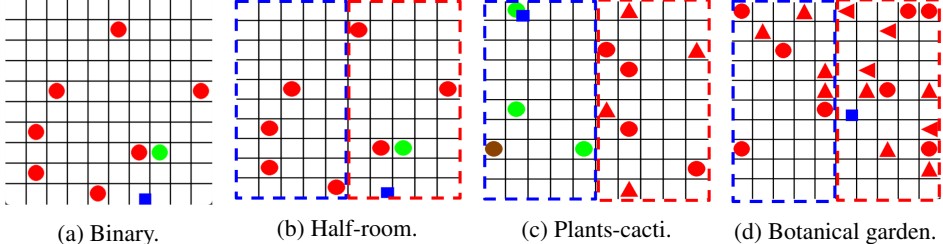

| (a) Binary. | (b) Half-room. | (c) Plants-cacti. | (d) Botanical garden. |

Figure 2: Environments frame.

to estimate the Q-value for all joint action pairs $(a^E, a^M)$; and ii) instead of using the environment reward, DQN utilizes the predicted reward provided by the reward model. Consequently, both the reward model and Q-network are trained simultaneously. Nonetheless, we assume that observed rewards are truthful which allows the reward model to converge under these conditions.

Alongside the reward model, we evaluate two baselines that treat Mon-MDPs as traditional MDPs first explored in the tabular setting (Parisi et al., 2024b): i) $\perp = 0$, undefined rewards are replaced with 0 when updating the Q-network; and ii) "ignore", updates the Q-network only with samples where rewards are observable, discarding samples with unobservable rewards. We aim to show that these limitations persist even when using FA.

In Mon-MDPs, the success of reward model generalization depends critically on the: i) state representation; ii) generalizing ability of the function approximator, and iii) observability of the reward in the environment. When these things do not align, we call this phenomenon "overgeneralization", as the following experiment section will demonstrate.

This work is not aiming to benchmark Deep RL algorithms, such as DQN, on Mon-MDPs, nor is it claiming as to which algorithm is best suited to this setting. The goal is to explore the effect of reward models when rewards are not always observed, and do so outside the tabular setting. The choice of DQN for this study is motivated primarily by its well-known and well-understood properties, focusing the conclusions on the role of FA.

## 4 EXPERIMENTS AND RESULTS

This section presents a series of experiments in non-tabular Mon-MDP settings, evaluating the capabilities and limitations of FA. All experiments are based on the plant-watering robot example introduced as a motivation example in Section 1.

The **plant-watering environment** is a $10 \times 10$ grid-world (see Figure 2a), where the agent, represented by a blue square, has six discrete actions: $\{\uparrow, \downarrow, \rightarrow, \leftarrow, Water, Stay\}$. The agent aims to maintain the hydration of eight plants, represented as circles. Each plant has three levels of dryness {fully dry (red), partially dry (brown), wet (green)}. Agent observations consist of six channels that capture the agent's location, plant type (three channels), dryness level, and walls. The agent perceives the environment through an egocentric-view with a window size $= 11$, where the agent is in the center of the window, resulting in state dimensions of $(6 \times 11 \times 11)$. Consequently, this window size does not always provide a complete view of the environment unless the agent is centered in the grid. This high-dimensional representation captures spatial and contextual information, increasing the complexity of learning compared to traditional tabular environments.

At the beginning of each episode, the agent's location and the plant positions are initialized randomly without replacement to ensure variability across episodes. All plants are initialized fully dry. The agent receives an environment reward of $r^E = +1$ for watering a fully or partially dry plant, and the plant's dryness level decreases by one unit. However, if the agent waters a wet plant, it will receive a penalty of $r^E = -1$. Moreover, the agent will receive a small penalty of $r^E = -0.2$ for watering an empty cell, effectively spilling water on the floor. Therefore, the agent's goal is to water plants appropriately while avoiding overwatering. A stochastic drying process causes some plants to become dry. At each timestep, with a probability defined by the dryness rate $= 5\%$, a randomly selected plant's dryness level increases by one unit, reflecting natural dehydration over time. This environment has no termination condition, but the episode is truncated after $100$ timesteps[4].

---

[4]We keep our experiments simple to isolate the generalization challenges of Mon-MDPs and FA.

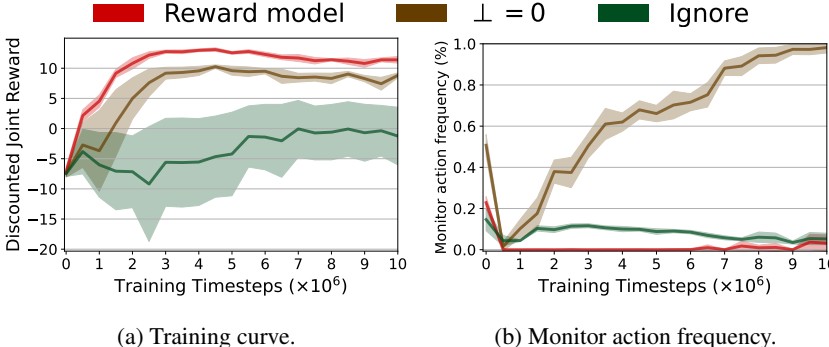

(a) Training curve.  (b) Monitor action frequency.

Figure 3: Reward model, "ignore", and $\perp = 0$ performance in the binary environment. The bold line represents the mean over 10 seeds; the shaded area is a $95\%$ confidence interval.

## 4.1 BINARY MONITOR

The first experiment tests whether combining FA with a learned reward model can achieve near-optimal policies in Mon-MDPs beyond tabular settings, whereas other baselines fail even when using FA (Contribution 1). This experiment introduces a binary variant of the plant-watering environment, shown in Figure 2a, where the agent has two additional monitoring actions alongside the six environment actions, $a^M \in \{$ask to be monitored, not ask to be monitored$\}$. If the agent asks to be monitored, it observes the true environment reward $\hat{r}^E = r^E$ and receives a monitoring reward $r^M = -0.2$ on that step. Choosing not to be monitored results in unobservable rewards $\hat{r}^E = \perp$ but avoids the monitoring penalty $r^M = 0$. Therefore, the optimal policy should be to water all plants while avoiding over-watering, spilling water on the floor, or requesting monitoring to avoid incurring a penalty. However, the agent will first need to request monitoring to learn such a policy.

Although the binary environment appears relatively simple, modeling it as a tabular environment requires representing more than $10^{18}$ distinct states. Consequently, storing the corresponding reward model and Q-value table requires an excessively large amount of memory. This challenge emphasizes the critical need for FA, which enables agents to generalize across similar states.

For a fair comparison, all algorithms utilize the same DQN architecture for the reward model, $\perp = 0$, and "ignore", and hyper-parameters are tuned for all methods individually, including the learning rate for the reward model ($\eta^R$), the Q-network ($\eta^Q$), and the $\epsilon$-decay rate (see Appendix A.1).

Figure 3a illustrates the training curve in the binary environment; the reward model outperforms $\perp = 0$ and "ignore" approaches, with a negligible variation across seeds. Complementing this, Figure 3b provides a quantitative analysis of monitoring frequency for each policy during training. The reward model rapidly learns to stop requesting monitoring, whereas $\perp = 0$ always needs to be monitored. Interestingly, "ignore" selectively requests monitoring only when watering plants, as it only learns from samples where rewards are observable. These results demonstrate that when employing the reward model, the learned policy empirically converges to the optimal policy in the binary environment. Furthermore, other methods that treat Mon-MDPs as traditional MDPs converge to sub-optimal policies. Specifically, $\perp = 0$ converges to ask continuously to be monitored, and "ignore" learns to ask to be monitored when watering plants. This highlights the reward model's ability to generalize effectively in non-tabular environments, addressing Contribution 1.

## 4.2 HALF-ROOM ENVIRONMENT

The half-room experiment investigates Contribution 2, evaluating whether an agent can generalize from monitored states with observable rewards to unmonitored states with unobservable rewards. To replicate this scenario, we adopt the plant-watering environment, introducing the half-room environment. In this setup, the agent does not have a monitoring action as part of its action space. Monitoring is automatically applied only in the left-half of the room, as illustrated in Figure 2b. In the monitored region (Zone 1), the monitor state is active ("on") $\Rightarrow s^M = 1$, and the proxy reward equals the true environment reward $\hat{r}^E = r^E$ and $r^M = 0$. However, when the agent moves to the right-half of the room (Zone 2), the monitor state is inactive ("off") $\Rightarrow s^M = 0$, in this region, the proxy reward is unobservable $\hat{r}^E = \perp$ and $r^M = 0$. The half-room environment mimics a scenario in which monitoring (*e.g.*, from human observation or monitoring sensors) is only available in a

specific part of the environment. In this setup, the agent must learn to generalize its behavior to the unmonitored region with unobservable rewards. Consequently, the optimal policy in the half-room environment involves properly watering plants, regardless of location.

In this experiment, the monitor state is represented by a one-hot encoding $\{0, 1\}$, indicating whether a given state is monitored, and the reward model is trained exclusively on samples from Zone 1. To integrate monitor awareness into decision-making, the Q-network processes the environment state through two convolutional layers to extract features. These features are then concatenated with the one-hot encoding of the monitor state and passed through the last two fully connected layers, estimating the Q-value for each action. By leveraging the neural network capacity for generalization, the reward model generalizes from the monitored section (Zone 1) to the unmonitored section (Zone 2). This generalization is evidenced by the agent's ability to appropriately water plants in both zones, despite only receiving reward signals from Zone 1. Since both zones contain an equal number of plants, comparing the training rewards reveals that rewards in both zones are comparable, indicating that the reward model not only learns in the monitored region but can also transfer its learned behavior to the unmonitored region, as shown in Figure 4a. These results indicate effective generalization[5]. However, a negligible difference in rewards exists between the two zones, which can be attributed to the presence of walls surrounding the room, given that some environment states in Zone 1 appear visually different from those in Zone 2.

This ability to generalize is significant when considering the broader implications for Mon-MDPs. According to Parisi et al. (2024b), the half-room environment is defined as an unsolvable Mon-MDP because the agent cannot observe rewards for nearly half of the environment states. Thus, the agent cannot distinguish between two Mon-MDPs that have different optimal policies, *e.g.*, whether plants should be watered on the right side of the room. Therefore, we demonstrate that by combining a reward model with FA, the agent can infer unobserved rewards to learn a near-optimal policy. This result highlights the potential of FA in overcoming partially observable rewards, enabling agents to effectively navigate environments that would otherwise be unsolvable, supporting Contribution 2.

### 4.3 PLANT-CACTUS ENVIRONMENT

This experiment explores a key limitation of FA in unsolvable Mon-MDPs: *overgeneralization*, where agents incorrectly extrapolate rewards from monitored to unmonitored environment states, resulting in suboptimal or undesirable behaviors (Contribution 3). We establish a formal definition of *overgeneralization* as the ratio between the reward model error on the unobservable of state-action pairs, $\mathbb{E}_{(s^E, a^E)}[(\hat{\mathcal{R}}(s^E, a^E, \theta) - r_t^E)^2 | \hat{r}_t = \perp]$, to the observable ones, $\mathbb{E}_{(s^E, a^E)}[(\hat{\mathcal{R}}(s^E, a^E, \theta) - r_t^E)^2 | \hat{r}_t = r_t^E]$. A ratio less than one indicates underfitting; a ratio near one (as in the binary environment) indicates effective generalization; a ratio much greater than one indicates the reward model fails to generalize well to unmonitored states. This notion is very similar to Witty and colleague's (2018) notion of extrapolation performance with respect to unreachable states for traditional MDPs.

To empirically study *overgeneralization*, we introduce a new plant: *cacti*. As illustrated in Figure 2c, the monitored zone contains four plants, while the unmonitored zone includes four plants and four cacti (triangle pointing up). Cacti differ from plants in their state tensor representation, ($[0, 1, 1]$) compared to ($[1, 1, 0]$), and in dynamics, since watering them incurs a penalty of $-1$. Crucially, since cacti appear only in the unmonitored zone, the agent never directly observes this penalty.

Figure 4b presents watering frequency for plants, cacti, and the floor in both the monitored and unmonitored zones throughout training. Initially, the agent spills water on the floor, but quickly learns to avoid this due to the penalty. Similar to the half-room environment, the agent waters plants in both zones. However, despite their distinct state representation, it also waters cacti in the unmonitored zone at a similar rate. This result demonstrates that FA can lead to incorrect reward extrapolation, causing the agent to adopt undesirable behaviors (*e.g.*, watering cacti). This highlights the challenge outlined in Contribution 3 and underscores the need for strategies to mitigate overgeneralization in Mon-MDPs.

---

[5]We excluded the $\perp = 0$ and "ignore" baselines from Figure 4 to keep the Figure readable. For completeness, a comparative figure including these approaches is provided in the Appendix A.2.

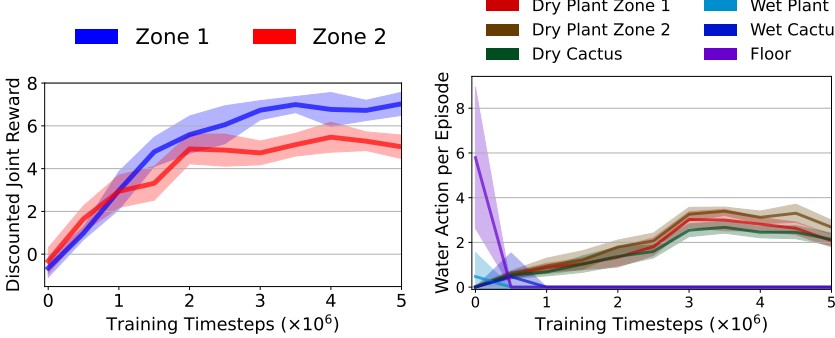

(a) Training curve per zone in the half-room.  (b) Watering frequency in the plant-cactus.

Figure 4: Results for the half-room and the plant-cactus environments. The bold line represents the mean over 10 different seeds; the shaded area is a 95% confidence interval.

## 4.4 BOTANICAL GARDEN ENVIRONMENT

Building on insights from the plant-cactus environment, where the reward model with FA exhibited a sub-optimal policy by watering cacti, the botanical garden environment addresses Contribution 3, evaluating whether agents can learn to act cautiously when encountering novel states, specifically, exploring how agents treat novel plants in the unmonitored zone. Overgeneralization is directly linked to epistemic uncertainty, which arises when the model extrapolates to unmonitored state-actions. This uncertainty can be quantified by the variance across ensemble models: $\mathbb{E}_{(s^E, a^E)}[\text{Var}_{i \in [1,N]}(\hat{\mathcal{R}}^i(s^E, a^E, \theta))|\hat{r}_t = \bot]$, where $\hat{\mathcal{R}}^i(s^E, a^E, \theta)$ is the $i$-th reward model out of $N$ reward models in the ensemble.

In the botanical garden environment, the monitored zone contains four standard plants and four cacti, allowing the agent to observe the outcomes of watering both. In contrast, the unmonitored zone includes four standard plants, four cacti, and four novel plants represented with a left-pointing triangle, as shown in Figure 2d. The environment encodes plant types using three channels as follows: floor $[0,0,0]$, plants $[1,1,0]$, cacti $[0,1,1]$, and novel plants are sampled from $\{[0,0,1], [0,1,0], [1,0,0], [1,0,1], [1,1,1]\}$.

To explore the cautious behavior in the unmonitored zone, we adapt the learning to be cautious framework (Mohammedalamen et al., 2021) to Mon-MDPs. First, an ensemble of 500 reward models is trained on samples from the monitored zone, with each reward model initialized randomly to capture reward uncertainty. Second, we optimize an approximation of CVaR using $k$-of-$N$ CFR (Chen & Bowling, 2012), where the $k/n$ ratio determines the robustness level, ranging from risk-neutral $k = N$ to highly risk-averse $k = 1$ and $N \gg k$. Our evaluation considers a range of policies, including that optimized from the original reward model, a mid-level robust policy 5-of-10, and a more robust policy 1-of-10, focusing on the watering frequency for standard and novel plants across both zones.

Table 1 reports the mean and 95% bootstrapped confidence interval (estimated by resampling runs with replacement and computing the mean) of watering frequency per episode for the reward model. It also reports the watering frequency ratio relative to the reward model for robust policies 5-of-10 and 1-of-10. The reward model exhibits a higher watering frequency for standard plants in the unmonitored zone than in the monitored zone. It also waters some novel plants $[0,1,0]$, $[1,0,0]$, and $[1,1,1]$. Interestingly, the reward model waters $[1,0,0]$ more frequently than standard plants in the monitored zone, likely because this representation is the inverse of cacti $[0,1,1]$, making it appear as a "super-plant". However, for other novel plants $[0,0,1]$ and $[1,0,1]$, the reward model never selects the watering. This suggests these representations inherently discourage watering.

In contrast, the robust policies 5-of-10 and 1-of-10, maintain a higher watering frequency for standard plants while significantly reducing watering for novel plants up to 5 times less than the reward model. Moreover, increasing the degree of robustness leads to a greater reduction in watering novel plants. These results indicate that robust policies balance persisting in rewarding actions that are highly familiar while mitigating overgeneralization in novel situations. Consequently, the findings support Contribution 3, demonstrating that learning a cautious policy mitigates overgeneralization.

Table 1: Watering frequency per episode in the botanical garden environment for the reward model and robust policies relative to the reward model. Results are reported as the mean and 95% confidence interval over 30 seeds. The last row shows novel plants that no policy has ever watered.

| Plant | Reward Model | Ratio 5-of-10 | Ratio 1-of-10 |
|---|---|---|---|
| Plants Zone 1 | 3.28 [3.00, 3.56] | ×1.30 [1.30, 1.30] | ×1.29 [1.29, 1.29] |
| Plants Zone 2 | 4.13 [3.80, 4.43] | ×1.17 [1.17, 1.17] | ×1.16 [1.16, 1.16] |
| [0, 1, 0] | 0.01 [0.00, 0.01] | ×0.27 [0.27, 0.29] | ×0.25 [0.25, 0.27] |
| [1, 0, 0] | 4.01 [0.81, 8.02] | ×0.20 [0.20, 0.23] | ×0.19 [0.19, 0.20] |
| [1, 1, 1] | 0.10 [0.06, 0.14] | ×0.56 [0.56, 0.58] | ×0.41 [0.41, 0.43] |
| [0, 0, 1], [1, 0, 1] | 0.00 [0.00, 0.00] | ×1.00 [1.00, 1.00] | ×1.00 [1.00, 1.00] |

## 5 Conclusion and Future Work

Although the Mon-MDP framework aims to model real-world applications where traditional MDPs fall short due to unobservable rewards, prior work has been limited to tabular environments with a finite, enumerable number of states. However, many real-world applications do not have finite state or action spaces or have an enormous number of states and actions. This work explores the challenges of non-tabular representations in Mon-MDPs and investigates function approximation capabilities and limitations. Our results demonstrate that training a reward model with FA allows agents to achieve near-optimal policies in some environments previously labeled as unsolvable. Additionally, we identify a critical limitation of FA in Mon-MDPs: *overgeneralization*, where the model incorrectly extrapolates rewards, leading to suboptimal and potentially unsafe behaviors. To mitigate this, we adopt a robust policy optimization approach for Mon-MDPs, demonstrating that agents can learn to act cautiously, even when rewards are unobservable. Although our approach builds on established tools (e.g., DQN, CVaR), the novelty lies in how these components integrate to address unobservable rewards in non-tabular settings while mitigating overgeneralization.

Moving forward, our research paves the way for multiple directions: i) extending Mon-MDPs to policy-based and actor-critic methods to handle continuous action spaces, including TRPO, PPO, and SAC (Konda & Tsitsiklis, 1999; Schulman et al., 2015; 2017; Haarnoja et al., 2018), ii) adopting computationally efficient approaches for capturing epistemic uncertainty, *e.g.*, noisy networks (Fortunato et al., 2018) and epistemic neural networks (Osband et al., 2023), iii) investigating the phenomenon of plasticity loss in Mon-MDPs when using deep neural networks, as we observed hints of that possibility in our experiments (see Appendix A.4), and exploring whether existing mitigation strategies in traditional MDPs can be effectively adapted to Mon-MDPs (Abbas et al., 2023; Nikishin et al., 2023; Dohare et al., 2024; Elsayed & Mahmood, 2024), iv) validating Mon-MDPs in real-world applications, such as industrial automation and autonomous robotics, to assess their practical feasibility and effectiveness, v) explore untruthful rewards (noisy or adversarial), and vi) conduct comprehensive risk-sensitive strategies comparison. Addressing these challenges will advance Mon-MDPs toward a robust framework for decision-making in complex, realistic environments with partially observable rewards, enabling more reliable and generalizable RL applications.

## 6 Ethics Statement

This work seeks to advance the field of RL by enabling the deployment of safe, reliable, and robust AI systems capable of navigating real-world complexities. Furthermore, this work sets the foundation for future research in settings where rewards are not always present. However, the community should be careful *not to solely rely* on such methods, as the automated safety measures are meant to complement, *not replace*, human judgment and safety planning.

## 7 Reproducibility Statement

The complete algorithm is described in Section 3, with pseudocode provided in Appendix Algorithm 1. Details of the experimental setup, including network architectures, hyper-parameters, computational resources, and environment descriptions, are provided in Appendix A. As part of the supplementary material, we include all relevant code and scripts, and the full codebase will be released publicly with the camera-ready version.

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

# Appendix

## A  EXPERIMENTS

---

**Algorithm 1** Learning to Be Cautious in Mon-MDPs

---

    **Initialize** $k$, $N$, number of iterations $T$, Q network $\hat{Q}(a^E, a^M | s^E, s^M)$.

    **Train** an ensemble of $N * T$ reward models $\hat{\mathcal{R}}(s^E, a^E, \theta)$ on samples from the monitored zone.

    **Input:** Environment & monitor state $(s^E, s^M)$.

    **Initialize:** The average over the least $k$ reward models $\hat{\mathcal{R}}_{k\text{-of-}N} = 0$.

    **for** $t = 0$ **to** $T$ **do**

        Sample $N$ reward models from the ensemble & get reward predictions $\hat{\mathcal{R}}_1(s^E), \cdots, \hat{\mathcal{R}}_n(s^E)$.

        Calculate the average over the least $k$ reward models $\hat{\mathcal{R}}_{k\text{-of-}N} \mathrel{+}= \frac{1}{k} \sum_{j=0}^{k} \hat{\mathcal{R}}_j(r|s^E)$.

    **end for**

    Update Q network using the average over the least $k$ reward prediction $\hat{\mathcal{R}}_{k\text{-of-}N}/T$.

---

Table 2: Notations table

| | |
|---|---|
| $\mathcal{S}$ | State Space. |
| $\mathcal{A}$ | Action Space. |
| $\mathcal{R}$ | Reward function. |
| $\mathcal{P}$ | Transition function. |
| $t$ | Timestep. |
| $\gamma$ | Discount factor. |
| $\pi$ | Policy. |
| $Q^\pi$ | Action-state-value for a policy $\pi$. |
| $\pi_*$ | Optimal policy. |
| $Q_*^\pi$ | Optimal action-state-value for a policy $\pi$. |
| $\theta$ | Function parameters. |
| $\hat{Q}^\pi$ | Approximate action-state-value for a policy $\pi$. |
| $\mathcal{S}^E$ | Environment state space. |
| $\mathcal{A}^E$ | Environment action space. |
| $\mathcal{R}^E$ | Environment reward function. |
| $\mathcal{P}^E$ | Environment transition function. |
| $\mathcal{M}$ | Monitor function. |
| $\mathcal{S}^M$ | Monitor state space. |
| $\mathcal{A}^M$ | Monitor action space. |
| $\mathcal{R}^M$ | Monitor reward function. |
| $\mathcal{P}^M$ | Monitor transition function. |
| $\hat{\mathcal{R}}$ | Predictive reward model. |
| $r^E$ | Environment reward. |
| $r^M$ | Monitor reward. |
| $\hat{r}^E$ | Proxy reward. |
| $\perp$ | Unobservable environment reward. |

Our neural network architecture, used for the reward model, $\perp = 0$, and "ignore" methods, consists of two convolutional layers followed by two fully connected layers, with rectified linear unit (ReLU) activations applied between them Glorot et al. (2011). The complete network architecture is detailed in Table 4, and hyper-parameters for all our experiments are summarized in Table 3.

The networks are trained using the Adam optimizer Kingma & Ba (2015) to minimize the mean squared error (MSE), where the reward model aims to minimize the error between predicted and actual rewards, while Q-models minimize the error between estimated and actual discounted returns.

Our code is provided as a ZIP file included in the supplementary materials. The `code` directory contains scripts for training reward models, Q-networks, and ensembles of reward models, as well as for performing $k$of-$N$ CFR optimization using PyTorch Paszke et al. (2017).

All our agents and models were trained on a 2.65GHz AMD® EPYC™ 7413 (Zen 2) CPU with 50 GB of memory, with access to $\frac{1}{5}$ of an Nvidia® A100SXM4™ GPU. Specifically, agents for the Binary, Half-room, and Plant-cactus environments were trained over a duration of 3 hours. While the Botanical garden environment, an ensemble of 500 reward models was trained, each requiring approximately 10 minutes, resulting in a total of 83.3 GPU hours. Additionally, 50 iterations of the $k$-of-$N$ CFR algorithm were executed on the CPU in under 15 seconds.

Table 3: Hyper-parameters values

| Hyper-parameter | Value |
| --- | --- |
| Discount factor $\gamma$ | 0.99 |
| Start training after | $10^5$ timesteps |
| Replay buffer size | $10^6$ samples |
| Target network update frequency | 50 |
| Batch size | 512 samples |
| Q-network update frequency | 250 episodes |
| Reward model learning rate ($\eta^R$) | $10^{-4}$ |
| Q-network learning rate ($\eta^Q$) | $10^{-4}$ |
| Initial exploration rate $\epsilon$ | 1.0 |
| Minimum exploration rate $\epsilon$ | 0.1 |
| Exploration $\epsilon$ linear decay rate | $10^{-4}$ |
| Weight decay | 0.0 |
| Optimizer | Adam Kingma & Ba (2015) |

Table 4: Network architecture

| | |
| --- | --- |
| First convolution layer kernel size | 5 |
| First convolution layer stride | 1 |
| First convolution layer number of channels | 32 |
| Second convolution layer kernel size | 3 |
| Second convolution layer stride | 1 |
| Second convolution layer number of channels | 64 |
| Number of neurons in the hidden layer | 512 |

A.1 HYPER-PARAMETERS TUNING

In addition to the reward model, we evaluate two alternative methods that treat Mon-MDPs as traditional MDPs:

1. $\perp = 0$, undefined rewards are replaced with 0 when updating the Q-network.

2. "Ignore", updates the Q-network only with samples where rewards are observable, discarding samples with unobservable rewards.

For a fair comparison, all algorithms employ the same DQN architecture as the reward model. The learning rate for both the reward model and Q-network, denoted as $\eta^R, \eta^Q$, is set to 0.0001. Additional experiments are conducted using alternative learning rates of (0.0005, 0.001, and 0.005). The area under the training curve for each setting is summarized in Table 5, with the mean and a 95% confidence interval reported.

Exploration follows an $\epsilon$-greedy strategy, where $\epsilon$ starts at 1.0 and linearly decays to 0.1 over time. In the binary environment, we tune the decay rate as illustrated in Figure 5a. Similarly, Figure 5b illustrates the tuning of the exploration decay rate for the half-room environment. Both figures compare different exploration decay rates $10^{-4}, 5 * 10^{-4}, 10^{-3}, 5 * 10^{-3}$. In both environments, smaller

exploration decay rates lead to a lower area under due to an increased number of exploratory steps. However, this additional exploration does not provide significant benefits, as both environments are relatively simple to explore.

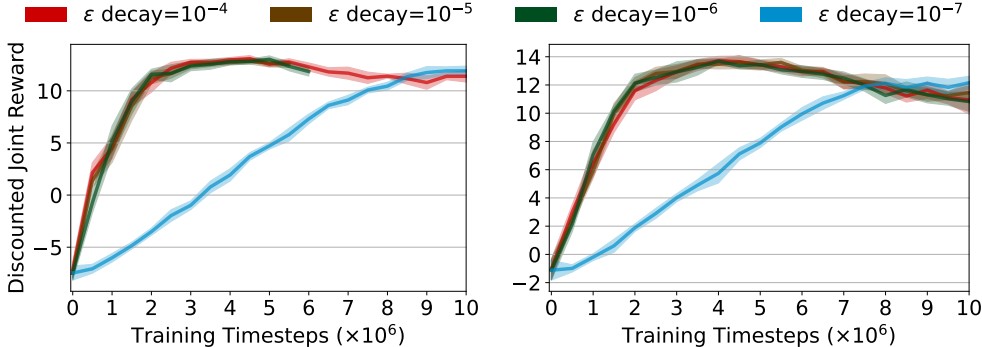

(a) Training curve for different exploration $\epsilon$ decay rates in the binary environment.

(b) Training curve for different exploration $\epsilon$ decay rates in the half-room environment.

Figure 5: Exploration $\epsilon$ decay rate tuning in the binary and half-room environments. The bold line represents the mean over 10 seeds; the shaded area is a 95% confidence interval.

Table 5: Area under the training curve for the reward model: hyper-parameters tuning in the binary environment, reporting the mean and a 95% confidence interval.

| $\eta^Q$ | $\eta^R$ | Area under the training curve |
|---|---|---|
| $10^{-4}$ | $10^{-4}$ | $20.00\ [19.81, 20.18]$ |
| $10^{-4}$ | $5*10^{-4}$ | $19.79\ [19.62, 19.96]$ |
| $10^{-4}$ | $10^{-3}$ | $19.69\ [19.32, 20.04]$ |
| $10^{-4}$ | $5*10^{-3}$ | $19.58\ [19.34, 19.82]$ |
| $5*10^{-4}$ | $10^{-4}$ | $19.35\ [19.18, 19.54]$ |
| $5*10^{-4}$ | $5*10^{-4}$ | $19.54\ [19.34, 19.76]$ |
| $5*10^{-4}$ | $10^{-3}$ | $19.27\ [19.03, 19.51]$ |
| $5*10^{-4}$ | $5*10^{-3}$ | $19.20\ [18.94, 19.48]$ |
| $10^{-3}$ | $10^{-4}$ | $18.69\ [18.38, 18.95]$ |
| $10^{-3}$ | $5*10^{-4}$ | $18.92\ [18.64, 19.14]$ |
| $10^{-3}$ | $10^{-3}$ | $18.45\ [18.16, 18.80]$ |
| $10^{-3}$ | $5*10^{-3}$ | $17.62\ [14.76, 19.17]$ |
| $5*10^{-3}$ | $10^{-4}$ | $14.04\ [10.94, 15.82]$ |
| $5*10^{-3}$ | $5*10^{-4}$ | $15.36\ [14.63, 16.14]$ |
| $5*10^{-3}$ | $10^{-3}$ | $11.25\ [6.58, 14.75]$ |
| $5*10^{-3}$ | $5*10^{-3}$ | $5.51\ [1.11, 9.96]$ |

## A.2 REWARD MODEL, "IGNORE" AND $\perp = 0$ PERFORMANCE

In the binary environment, in addition to the reward model, we evaluate $\perp = 0$, replacing undefined rewards with $0$, and "ignore" updating the Q-network only with samples where rewards are observable. For a fair comparison, all algorithms utilize the same DQN architecture as the reward model, and performance is reported as the mean with a 95% confidence interval. This is illustrated in Figure 6, where the 95% confidence interval is represented as a shaded area, and in Figure 7, which shows the individual runs.

A quantitative comparison of the reward model against the "ignore" and $\perp = 0$ baselines in the half-room environment is presented in Figure 8. The $\perp = 0$ baseline exclusively waters the plants in the monitored side while completely neglecting the plants in the unmonitored side and frequently

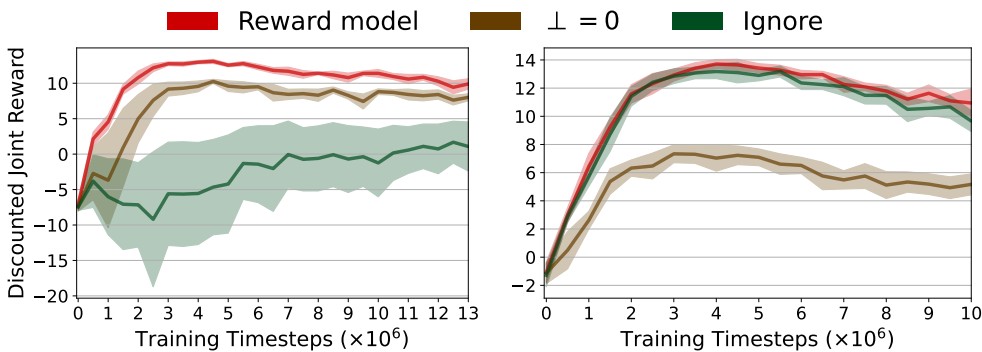

(a) Training curve in the binary environment.    (b) Training curve in the half-room environment.

Figure 6: Reward model, "ignore", and $\perp = 0$ training curves in the binary and half-room environments. The bold line represents the mean over 10 seeds; the shaded area is a $95\%$ confidence interval.

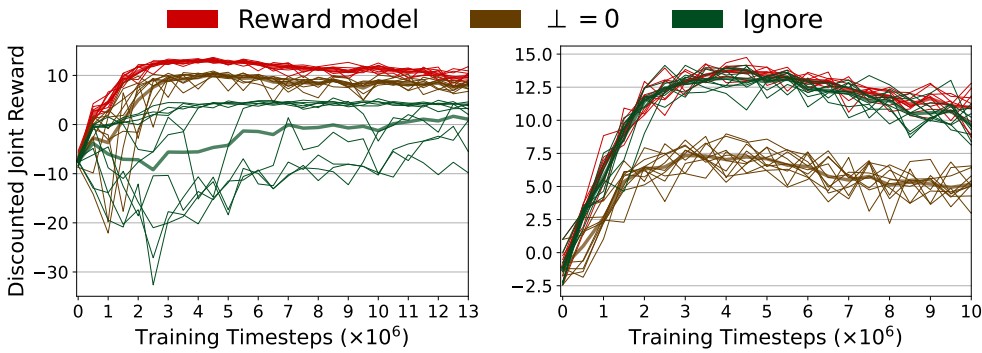

(a) Training curve in the binary environment.    (b) Training curve in the half-room environment.

Figure 7: Reward model, "ignore", and $\perp = 0$ training curves in the binary and half-room environments. The bold line represents the mean over 10 seeds, and each line represents a seed.

spills water on the floor. In contrast, "ignore" waters plants on the monitored side and partially on the unmonitored side, but avoids plants near walls. This is reflected by its lower reward in the unmonitored zone relative to the reward model.

Figure 9 shows a similar comparison in the plant–cactus environment. The $\perp = 0$ baseline again spills water on the floor and avoids watering both plants and cacti in the unmonitored zone. Meanwhile, "ignore" spills water on the floor more frequently and waters plants less consistently than the reward model.

Together, these results demonstrate that both "ignore" and $\perp = 0$ baselines exhibit suboptimal behavior in the half-room and plant–cactus environments.

## A.3    DETAILED RESULTS FOR THE BOTANICAL GARDEN ENVIRONMENT

The Botanical Garden environment serves as a benchmark for evaluating whether agents can learn to act more cautiously in such settings. Specifically, exploring how agents treat novel plants when no reward is observable (in the unmonitored zone).

Table 6 reports the average watering action frequency per episode for bother standard and novel plants, reporting the mean watering frequency per episode, along with $95\%$ bootstrapping confidence intervals, over 30 random seeds, for different approaches:

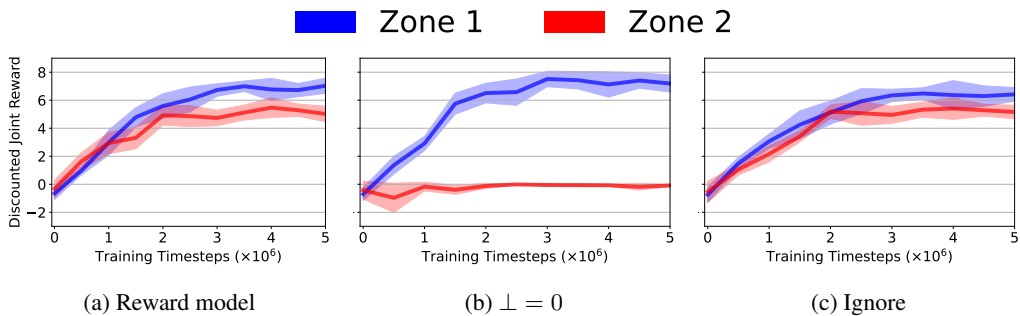

(a) Reward model          (b) $\perp = 0$          (c) Ignore

Figure 8: Reward model, "ignore", and $\perp = 0$ training reward per zone in the half-room environment. The bold line represents the mean over $10$ seeds; the shaded area is a $95\%$ confidence interval.

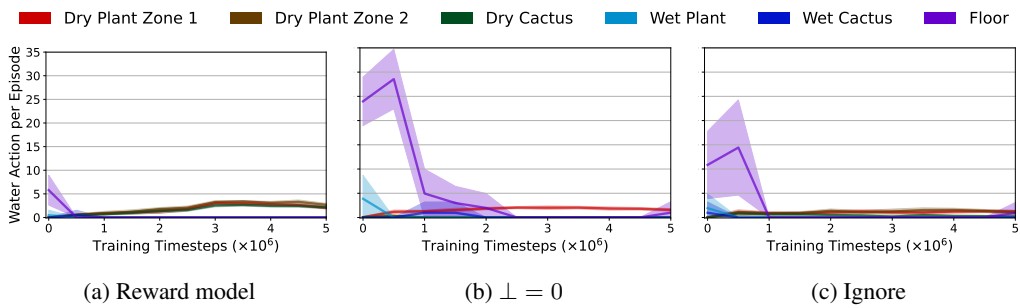

(a) Reward model          (b) $\perp = 0$          (c) Ignore

Figure 9: Reward model, "ignore", and $\perp = 0$ watering action frequency in the plant-cactus environment. The bold line represents the mean over $10$ seeds; the shaded area is a $95\%$ confidence interval.

1. The reward model-based policy relies solely on learned reward estimates.

2. The 10-of-10 policy considers the expected (average) outcomes across all sampled reward models, leading to a risk-neutral policy.

3. The 5-of-10 policy considers the worst $50\%$ of the reward models, leading to a mid-risk-averse policy.

4. The 1-of-10 policy considers the worst $10\%$ of the reward models, leading to a highly risk-averse policy.

In both zones, robust policies consistently increase watering frequency compared to the reward model. However, as the degree of robustness increases (more risk-averse), watering frequency decreases for standard plants in both zones. For novel plants in Zone 2 (unmonitored zone), robust policies further reduce watering frequency, with this effect becoming more evident as robustness increases. Interestingly, even the risk-neutral 10-of-10 policy, which optimizes for the expected distribution, exhibits some caution when dealing with novel plants. It waters them less frequently than the reward model, potentially leveraging epistemic uncertainty across all reward models. Notably, for certain novel plant states like $[0, 0, 1]$ and $[1, 0, 1]$, none of the policies apply watering actions, suggesting that these plant representations inherently discourage watering.

The ensemble of reward models provides a distribution that captures epistemic uncertainty. Figure 10 illustrates the histogram of predicted rewards for the "water" action. In the unmonitored region, the predictions from an ensemble of $500$ reward models exhibit low variance for input samples similar to those encountered during training (e.g., plant, cactus), indicating agreement. In contrast, for previously unseen samples (e.g., $[0, 0, 1]$), the ensemble exhibits a higher variance, reflecting increased epistemic uncertainty. These observations support our hypothesis that *overgeneralization* correlates with epistemic uncertainty arising from limited reward observability.

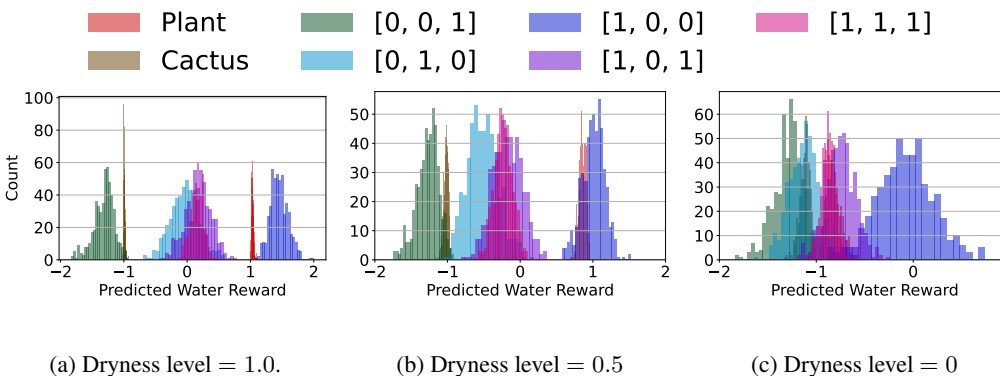

(a) Dryness level = 1.0.  (b) Dryness level = 0.5  (c) Dryness level = 0

Figure 10: Reward predictions histogram from an ensemble of 500 models for the "water" action, capturing the epistemic uncertainty.

Additionally, Table 7 presents the same results in a different format, showing the ratio of watering actions for robust policies 10-of-10, 5-of-10, and 1-of-10 relative to the reward model. This comparison highlights how each policy varies from the reward model in terms of watering frequency reduction.

Table 6: Water action frequency per episode for the reward model and robust policies 10-of-10, 5-of-10, and 1-of-10, in the botanical garden environment. Represented by the mean and 95% bootstrapped confidence interval over 30 seeds. The last row shows novel plants that have never been watered by any policy.

| Plant | Reward model | 10-of-10 | 5-of-10 | 1-of-10 |
|---|---|---|---|---|
| Plants Zone 1 | 3.28 [3.00, 3.56] | 4.78 [4.44, 5.09] | 4.27 [3.85, 4.68] | 4.22 [3.83, 4.55] |
| Plants Zone 2 | 4.13 [3.80, 4.43] | 5.45 [5.08, 5.78] | 4.83 [4.31, 5.24] | 4.77 [4.41, 5.14] |
| [0, 1, 0] | 0.01 [0.00, 0.01] | 0.00 [0.00, 0.00] | 0.00 [0.00, 0.00] | 0.00 [0.00, 0.00] |
| [1, 0, 0] | 4.01 [0.81, 8.02] | 1.23 [1.10, 1.37] | 0.82 [0.73, 0.90] | 0.75 [0.69, 0.80] |
| [1, 1, 1] | 0.10 [0.06, 0.14] | 0.10 [0.09, 0.11] | 0.05 [0.05, 0.06] | 0.04 [0.03, 0.05] |
| [0, 0, 1], [1, 0, 1] | 0.00 [0.00, 0.00] | 0.00 [0.00, 0.00] | 0.00 [0.00, 0.00] | 0.00 [0.00, 0.00] |

Table 7: Water action frequency per episode for the reward model and the ratio for robust policies 10-of-10, 5-of-10, and 1-of-10 w.r.t the reward model, in the botanical garden environment. Represented by the mean and 95% bootstrapped confidence interval over 30 seeds. The last row shows novel plants that have never been watered by any policy.

| Plant | Reward model | Ratio 10-of-10 | Ratio 5-of-10 | Ratio 1-of-10 |
|---|---|---|---|---|
| Plants Zone 1 | 3.28 [3.00, 3.56] | ×1.46 [1.46, 1.46] | ×1.30 [1.30, 1.30] | ×1.29 [1.29, 1.29] |
| Plants Zone 2 | 4.13 [3.80, 4.43] | ×1.32 [1.32, 1.32] | ×1.17 [1.17, 1.17] | ×1.16 [1.16, 1.16] |
| [0, 1, 0] | 0.01 [0.00, 0.01] | ×0.40 [0.40, 0.41] | ×0.27 [0.27, 0.29] | ×0.25 [0.25, 0.27] |
| [1, 0, 0] | 4.01 [0.81, 8.02] | ×0.31 [0.31, 0.34] | ×0.20 [0.20, 0.23] | ×0.19 [0.19, 0.20] |
| [1, 1, 1] | 0.10 [0.06, 0.14] | ×0.99 [0.99, 1.04] | ×0.56 [0.56, 0.58] | ×0.41 [0.41, 0.43] |
| [0, 0, 1], [1, 0, 1] | 0.00 [0.00, 0.00] | ×1.00 [1.00, 1.00] | ×1.00 [1.00, 1.00] | ×1.00 [1.00, 1.00] |

## A.4 PLASTICITY LOSS IN MON-MDPS

Plasticity loss refers to the loss of flexibility in an agent's learning process as it encounters more samples. This phenomenon occurs when the agent becomes too committed to previously learned behaviors or strategies, preventing it from adapting effectively to new, unforeseen states or changes in the environment.

In Mon-MDPs, we hypothesize that plasticity loss is particularly noticeable because the agent learns two models: i) a reward model to overcome unobservable rewards, which influences the Q-network

by providing reward predictions, and ii) the Q-network itself, which updates its values based on both state and reward inputs.

Our experiments demonstrate this effect. In the binary environment, after 6M timesteps, the agent's performance deteriorates, as indicated by the drop in training reward in Figure 11. The agent even starts requesting to be monitored, further confirming its declining adaptability. Similarly, in the half-room environment, training beyond 5M-7M timesteps leads to a decline in accumulated rewards across both zones, as shown in Figure 12.

These findings suggest that the use of deep learning in Mon-MDPs may contribute to plasticity loss. As future work, we plan to investigate this phenomenon further and explore whether existing mitigation strategies from traditional MDPs (Abbas et al., 2023; Nikishin et al., 2023; Dohare et al., 2024; Elsayed & Mahmood, 2024) can be effectively adapted to Mon-MDPs.

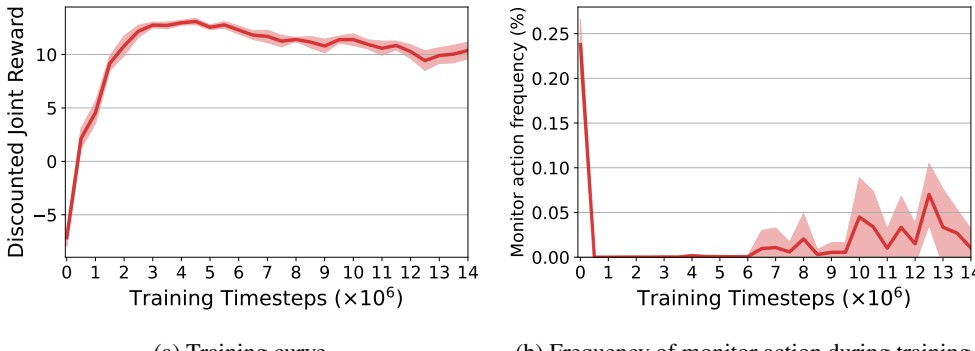

(a) Training curve.        (b) Frequency of monitor action during training.

Figure 11: Preliminary evidence for the loss of plasticity in the binary environment. The bold line represents the mean over 10 seeds; the shaded area is a 95% confidence interval.

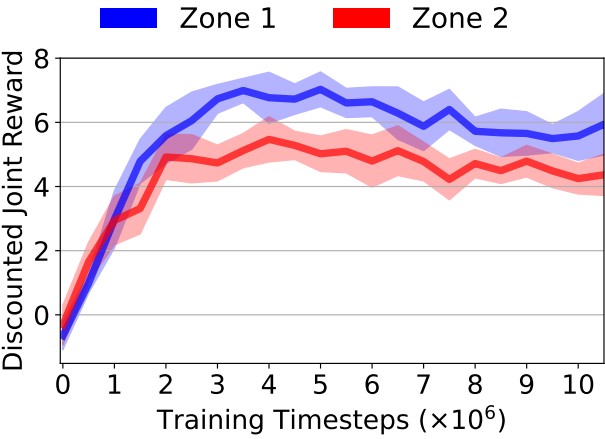

Figure 12: Preliminary evidence for the loss of plasticity in the half-room environment. The bold line represents the mean over 10 seeds; the shaded area is a 95% confidence interval.