# OpenReview forum: "Generalization in Monitored Markov Decision Processes (Mon-MDPs)"
_ICLR.cc/2026/Conference — Submitted to ICLR 2026_

### Official Review · Reviewer_PqK8 · 2025-10-29

**Soundness:** 2
**Presentation:** 2
**Contribution:** 2
**Rating:** 2
**Confidence:** 3

**Summary:**

This paper investigates the integration of function approximators into traditional tabular (monitored) MDPs to enhance scalability. The authors effectively demonstrate how value function approximators can improve an agent’s performance in various scenarios within a plant-watering environment. However, there are notable conceptual and methodological gaps in the proposed approach. In fact, the authors contradict several of their claims, as outlined in the weaknesses section.

**Strengths:**

•	Integrating traditional tabular MDPs with function approximators such as deep Q-networks.
•	Exploring several scenarios in the plant irrigation example.

**Weaknesses:**

•	There are multiple statements that sound contradictory to each other, which I point out in Question section.
•	In Section 4.3 (line 366), the authors claim the agent never observes the penalty. However, in the next paragraph, they state that the agent learns to avoid it. If the agent learns from the penalty, it must have been exposed to it, implying the overall reward is observable.
•	The authors state that traditional monitored MDPs operate in the tabular setting and that their function approximator generalizes the method to non-tabular settings. While this may hold theoretically, all presented experiments are still conducted in large but finite tabular domains. If the approach is intended for non-tabular MDPs, experiments in continuous state–action spaces should have been included.
•	The authors claim that traditional methods are limited to finite state and action spaces (Conclusion section). However, their own experiments are restricted to such finite spaces, which weakens this argument.
•	The paper asserts that the proposed approach is near-optimal, yet no theoretical guarantees or performance bounds are provided. Given that it is tested on a single application, its generalizability to other domains remains uncertain.

**Questions:**

1. In the main contributions on page 2, the authors claim that their approach can achieve near-optimal performance for unsolvable monitored MDPs. However, on page 3, they state that there is no optimal policy for unsolvable MDPs. If no optimal policy exists, it is unclear how their approach can be characterized as near-optimal.
2. In Section 3, the reward function is said to be trained on observable states. However, it is unclear how this trained model can accurately estimate rewards for unobservable states, particularly when their rewards differ substantially from those of observed states.
3. The reward function $r^E$ follows a simple deterministic rule in Section 4. How would your approach handle a stochastic reward function?
4. Is the optimal policy known in your examples? On what basis do you evaluate near-optimality?
5. For training the reward model, what happens when only a limited number of states are observable? Would the performance remain stable under sparse observability?
6. You mention robustness to overgeneralization. Could you clarify this? Does introducing a penalty bias the agent toward conservative, worst-case behavior?

---

> ### Author Response · Authors · 2025-11-24
> **Authors' Response to Reviewer PqK8**
>
> We thank the reviewer for their feedback.
>
> ## Weakness:
> - **Contradictory statements**: In line 366, we stated that the agent never directly observes the environment penalty from watering cacti, because those cacti are located in the unmonitored zone. In contrast, the agent observes the reward for watering plants and the floor (spilling water), since there are examples in the monitored zone. However, in the following paragraph, we **never** mentioned that "the agent learns to avoid cacti", we explicitly said that "despite their distinct state representation, it also waters cacti in the unmonitored zone at a similar rate."
> - **Finite tabular experiments and non-tabular generalization claim**: Line 295 states that, representing the plant-watering environment more than $10^{18}$ distinct states, and storing the corresponding reward model and Q-value tables requires an excessively large amount of memory (around $10^{8}$ TB), highlighting the impracticality of tabular methods (intractable). For comparison, $19 \times 19$ Go board game could be represented as a tabular problem with around $10^{170}$ states. These numbers illustrate the motivation for using function approximation to handle extremely large state spaces.
> - **Continuous action space**: In Line 458, we stated that extending the reward model to a continuous action space is a future direction, and we have not said it is part of our contributions.
> - **No theoretical guarantees**: We agree that we do not provide theoretical guarantees; our claim of *near-optimal performance* is purely empirical and explicitly stated as such in environments where we know the optimal policy. The contribution of this work is to show that previously considered unsolvable Mon-MDPs outside tabular cases can still be handled effectively with function approximation. Extending these results to broader domains is an important direction for future work.
>
> ## Questions:
> - **Near-optimal policy in unsolvable Mon-MDPs**: The half-room environment is classified as an unsolvable Mon-MDP because the agent cannot observe rewards for nearly half of the states. While a uniquely optimal policy does not exist, some policies could be better than others. Our experiments demonstrate that by combining a reward model with function approximation, the agent can infer unobserved rewards and learn a policy that is near-optimal in practice, e.g., it waters plants in both regions (monitored and unmonitored).
> - **Accurate estimation rewards**: Function approximation is designed to capture features that correlate with the reward values. By learning these correlations, the model can generalize from observed states to unobserved states, allowing the agent to approximate rewards even when direct observation is unavailable.
> - **Stochastic reward function**: We would appreciate further clarification regarding the term ``stochastic''. Is the reviewer referring to random noise in the reward function, or adversarially generated rewards? Our current approach assumes a deterministic reward function, but we believe the framework could be extended to handle unbiased noise to the observable reward.
> - **Near optimal policy**: One reason for selecting intentionally simple environments is that the optimal policy is known. This allows us to quantitatively measure near-optimality as the difference between the learned policy’s performance and the known optimal performance.
> - **Sparse observability**: We acknowledge that sparse observability can challenge neural network-based reward models. We hypothesize that a sufficient number of samples is necessary to capture relevant features for generalization. Investigating how sample size affects performance and generalization remains an important future direction.
> -  **Quantitative measure of cautious behavior**: We quantify cautiousness by metrics such as the number of times the agent waters different plants or takes actions in potentially unsafe states.

---

### Official Review · Reviewer_AuFX · 2025-11-01

**Soundness:** 3
**Presentation:** 3
**Contribution:** 3
**Rating:** 6
**Confidence:** 2

**Summary:**

This paper extends the Monitored MDP framework (Parisi et al., AAMAS 2024) beyond the tabular regime into function approximation settings, thereby improving its practical applicability. The framework considered in this paper allows partially unobservable rewards. For this, the paper proposes to combine a neural reward model based on DQN function approximation. The paper empirically investigates the proposed approach. The authors found that function approximation with reward modeling may recover near-optimal policies even under unobserved rewards. In addition, interestingly, they also discovered that function approximation may induce an issue of overgeneralization, where agents incorrectly extrapolate rewards. Motivated by this, they extend the "learning to be cautious" originally designed for traditional MDPs to the setting of Mon MDPs.

**Strengths:**

- The paper extends the concept of Mon-MDPs to practically relevant settings based on function approximation, e.g., DQN.
- The paper discovers the issue of overgeneralization when applying the framework. It provides a quantitative definition of overgeneralization and analyzes it based on numerical experiments.
- The paper establishes empirical demonstrations that combine function approximation and reward modeling. The result is very interesting in that the framework can find a near-optimal policy even for instances that are formally considered to be unsolvable.
- Experiments are well-designed, and risk-averse RL, along with the "learning to be cautious" techniques, are carefully adopted.

**Weaknesses:**

- The paper has no new theoretical results, although it may not be the main scope of this paper.
- Experiments are illustrative, but they are confined to grid-worlds. Perhaps further validations in continuous or high-dimensional control domains would be desired. However, to be fair, the numerical experiments consider instances with a larger scale, in comparison with the AAMAS paper.

**Questions:**

-

---

> ### Author Response · Authors · 2025-11-24
> **Authors' Response to Reviewer AuFX**
>
> We thank the reviewer for the thoughtful and encouraging feedback. We appreciate the recognition of our contributions.
>
> ## Weakness:
> - **Lack of theoretical results**: We agree that the paper does not introduce new theory; our primary aim was to empirically examine generalization and reward observability in non-tabular Mon-MDPs. We view our empirical findings as a necessary precursor to future theoretical developments.
> - **Experiments limited to grid-worlds**: Our choice of plant-watering environment was intentional to isolate the core challenges of generalization and reward observability without confounding factors. Extending Mon-MDPs to high-dimensional control tasks is an exciting and important future direction.

---

### Official Review · Reviewer_5Zm1 · 2025-11-01

**Soundness:** 3
**Presentation:** 2
**Contribution:** 2
**Rating:** 4
**Confidence:** 3

**Summary:**

The paper studies Monitored MDPs (Mon-MDPs), where the environment is endowed with a separate monitoring (also a MDP) process
The prior work only dealt with tabular settings of the Mon-MDPs; this paper extends prior tabular treatments by pairing function approximation (FA) with a learned reward model. The paper has a number of simple-to-explain experiments to show generalization (i.e. positive effect of reward extrapolation to unmonitored regions of the environment ) and also a failure mode (where rewards when extrapolated to unmonitored regions lead to unintended consequences). As a remedy, they propose risk-averse policy optimization to act cautiously.

**Strengths:**

1) The Mon-MDP setting is clearly defined. Although I do not clearly follow the impact, utility of the same specially in the example and experiments of the authors (more on this below)

2) The set of experiments ,while preliminary ,seems to be a good starting point in exploring this line of research

3)The paper is well written in the sense, at many places it explains the scope of its results and settings and does not seem to overclaim results.

**Weaknesses:**

1) Disconnect between the model and the experiments

While the Mon-MDP formalism is general, the experiments instantiate only degenerate monitors, either deterministic spatial gating of observability or a binary ask/no-ask with fixed cost. These settings could be modeled without a stateful monitor MDP, and there are some papers that do just that. To justify the general model, I recommend adding scenarios with nontrivial monitor dynamics (at least giving examples where stateful monitor MDP makes sense, e.g., even the watering robot example in the introduction is vague and not clear what the monitor MDP is in that example) or, alternatively, narrowing the claims to the studied special cases.

2) Role of $r_M$ is very non-intuitive

Because the objective maximizes $r_𝑀+r_E$ , the monitor reward acts as an arbitrary scalarization of two objectives.
=0; the paper gives no guidance for selecting $r_M$. (e.g. Binary monitor example or no-monitor experiments can be really different with different $r_M$ selection, again in the watering robot example, what is $r_M$ ) . For now, I think it is just reflective ofthe  cost of observation or something similar

3) About Contribution 3

 The ‘overgeneralization’ result (watering cacti unseen during training) is unsurprising given that rewards for cacti are never revealed. As framed, it reads more like an instance of standard distribution shift than a novel Mon-MDP phenomenon. The paper does add a metric and a cautious policy mitigation, but the underlying failure case is expected. One way to circumvent it will be to make the failure non-obvious. e.g., create unmonitored states that are feature-similar to monitored ones (not a new class), show the ratio spikes anyway, then show the mitigation helps; or show a case where FA surprisingly doesn’t fail due to representation structure.


4)What is not tested is the reward misspecification under the mon-MDP setting. In all the experiments, whenever the agent can observe the $r_E$, it observes it accurately.

**Questions:**

1) Please see the weaknesses too

2) The parametrization is kind of implicit, i.e. architecture implicitly defines the function classes. It is unclear whether the results are artifacts of the architecture or the chosen parameterization. Can you confirm whether an architecture search or ablation study is required to provide evidence?

---

> ### Author Response · Authors · 2025-11-24
> **Authors' Response to Reviewer 5Zm1**
>
> We thank the reviewer for their feedback.
>
> ## Weakness:
> - **Disconnect between the model and the experiments**: While the Mon-MDP framework is general, our experiments intentionally span multiple distinct monitoring scenarios, including spatially selective reward observability, conditional querying with costs, and varying patterns of monitored vs. unmonitored states. These settings are not degenerate: they induce fundamentally different patterns of partial reward observability, which is precisely the phenomenon this paper investigates. In addition, as we show empirically, traditional MDP baselines fail, specifically because they cannot represent unobserved rewards, making Mon-MDPs essential rather than optional. Our goal in this paper is not to enumerate all possible monitor dynamics, but to study rewards observability and generalization. Exploring the full space of monitor designs is an important direction, but it is beyond the scope of this paper contribution.
> - **The role of $r^M$**: In Mon-MDP [1], the moniotor reward $r^M$ is a fundemntail pice, and it is not necessarily a cost or a penalty, decoupling the environment and the monitor rewards allows; i) clarifies what is observable versus unobservable, allowing agents to learn when and how reward signals are observable, ii) enables sophisticated exploration strategies, such as acting cautiously in unmonitored states or seeking feedback when monitoring is likely, and iii) supports transfer learning across tasks and domains, where either the monitor or the environment may change independently. Therefore, the monitor reward is not another penalty and could change depending on the monitoring scenario (e.g., $r^{M} > 0$ to encourage the agent to seek monitoring).
> - **Contribution 3**:
>     - Our overgeneralization definition (line 356) links it directly to the agent’s inability to observe rewards, which affects the agent's performance to maximize those unobservable rewards. In other words, overgeneralization shares some similarities with distribution shift as caused by epistemic uncertainty; however, it is different since the predictions will be fed to the agent's value function. In this sense, overgeneralization is unique to Mon-MDPs, not simply a general function approximation challenge.
>     - In fact, our environments already include feature-similar unmonitored states: the half-room (Section 4.2) setting contains identical plants with the same feature representation but placed on opposite sides of the room, where only one side is monitored. Complementarily, the plant–cacti environment tests generalization across different plant types (e.g., cacti), allowing us to study both within-class and out-of-class generalization under reward unobservability.
> - **Reward misspecification**: We would appreciate it if the review could elaborate more on what they mean by ``reward misspecification''. Is it unbiased noise to the observable reward $\hat{r}^E_t = r^E_t + \epsilon$, where $\epsilon$ is a Gaussian noise, or adversarial misspecification? In this paper (non-tabular setting), we give a minor relaxation of the truthful assumption; function approximation with an experience replay buffer could potentially cancel out unbiased Gaussian noise. Therefore, we will add the conditions to the paper.
> ## Questions:
> - **Network architectures**: Our architectural choices follow the standard configurations used in the original DQN work, which are widely adopted in the literature and serve as strong, well-understood baselines. The goal of this paper is not to study architectural effects, but to isolate the phenomena that arise from reward unobservability in non-tabular Mon-MDPs. While different architectures may vary in representational capacity, the key behaviors we report stem from the interaction between Mon-MDP structure and function approximation, not from specific network design choices.
>
> ## References:
> [1] Parisi, Simone, et al. "Monitored Markov Decision Processes." Proceedings of the 23rd International Conference on Autonomous Agents and Multiagent Systems. 2024

---

> > ### Comment · Reviewer_5Zm1 · 2025-11-25
> > **thanks for your rebutal**
> >
> > Thanks for carefully answering my questions.
> > By reward misspefication, I meant that either the rewards are noisy or adversarial. I think this relates to my point that Mon-MDPs, as described, are very general, while the experiments only test a specific class/type of Mon-MDP. In the general version of Mon MDP, as the authors write, the observed rewards may differ significantly from the actual rewards. I was trying to hint towards this point.

---

> > > ### Author Response · Authors · 2025-11-28
> > > **Authors' Response to Reviewer 5Zm1**
> > >
> > > We thank the reviewer for the prompt response and for clarifying the meaning of **reward misspecification**, and for acknowledging that our earlier answers addressed the previous questions.
> > > - We agree that the Mon-MDP framework allows for a wide spectrum of monitoring behaviors, including cases where the observed reward may be noisy, biased, or even adversarial to the true environment reward. However, in this paper, experiments are restricted to truthful rewards. This design choice allows us to isolate and study reward observability and generalization without the additional confounding effects introduced by untruthful rewards.
> > > - We **revised the paper** to clearly state the assumptions made about the reward monitoring process in our experiments, e.g., observed rewards are truthful. In addition, discuss untruthful rewards (noisy or adversarial) as an important future work.

---

### Official Review · Reviewer_5gM9 · 2025-11-03

**Soundness:** 2
**Presentation:** 3
**Contribution:** 1
**Rating:** 2
**Confidence:** 4

**Summary:**

The paper investigates the use of nonlinear function approximation coupled with a learned reward model for monitored Markov decision processes (Mon-MDPs). The authors argue that their approach improves generalization, as it allows agents to generalize from monitored states with observable rewards to unmonitored states with unobservable rewards. The caveat is that agents may sometimes overgeneralize, as they inaccurately extrapolate rewards. The authors corroborate these findings with empirical evaluation

**Strengths:**

- The Mon-MDP framework is relatively recent, and has not been investigated in adequate depth. This paper makes a first effort to shed light on some critical challenges and limitations of Mon-MDPs, while also discussing ways to mitigate them (e.g., by learning a reward model).
- The experimental study contains targeted experiments that are able to confirm the authors' findings.

**Weaknesses:**

- In my view, the novelty is limited. The authors essentially explore the use of function approximation and a learned reward model for Mon-MDPs. This is a straightforward idea without substantial innovation. Function approximation has been known for a long time to improve the RL performance and is nowadays an almost indispensable component of modern reinforcement learning systems. Adding it to Mon-MDPs makes a lot of sense, but is otherwise straightforward. Likewise, the learned reward model makes a lot of sense in this setting, but it's otherwise not a substantial innovation.
- There is no theory, e.g., with respect to the learned reward model. The authors try to corroborate their findings through their empirical evaluation, but the paper lacks formal theory.
- The experiments mainly study relatively simple MDPs. It was not clear why the authors did not try more complex MDPs. Perhaps the Mon-MDP setting is not so broad?
- The mitigation strategy discussed in the paper for overgeneralization is in my view one of the most interesting parts of this work, but it is only covered rather superficially.

**Questions:**

- Have the authors considered more complex MDPs, where they may even need to apply more involved RL algorithms? Perhaps somehow more realistic MDPS?
- The authors could have explored mitigation strategies in more detail. No details for instance are provided regarding CVaR and k-of-N CFR. Why the authors decide to use an approximation of CVaR and not the actual CVaR, and also what motivated their use of k-of-N CFR? None of that is covered in the paper. I personally feel the paper would have benefited a lot from a more careful treatment of the phenomenon of overgeneralization and the possible mitigation strategies.

---

> ### Author Response · Authors · 2025-11-24
> **Authors' Response to Reviewer 5gM9**
>
> We thank the reviewer for their feedback.
> ## Weakness:
> - **Novelty**: The core contribution of this work is not the use of function approximation itself, but the identification and empirical characterization of new challenges that arise uniquely when function approximation interacts with Mon-MDPs, including: observability, cautiousness, and performance in unmonitored regions. These phenomena do not occur in traditional MDPs, even with function approximation, and have not been documented or analyzed in prior work.
> - **lack of theory**: While we agree that developing formal guarantees for learned reward models in Mon-MDPs is an important future direction, establishing such theory requires first understanding the empirical phenomena and failure modes, which this paper provides for the first time. Our work lays the foundation for theoretical analyses by revealing when and how function approximation-based reward models succeed or fail under partial reward observability.
> - **Environment complexity**: We intentionally focus on simple Mon-MDPs to isolate the interaction between generalization and reward observability. More complex environments introduce confounding factors such as stochastic transitions, which would obscure the core phenomena we aim to study. The simplicity is thus methodological, not a limitation of the Mon-MDP framework. The Mon-MDP framework is a general framework that captures settings where rewards are not always observable (including the traditional MDP).
> - **Overgeneralization**: We agree that the mitigation strategy is an important contribution. Our goal in this paper is to introduce, diagnose, and mitigate overgeneralization; a full exploration of mitigation is substantial enough to merit a dedicated follow-up study. Even so, we provide initial evidence that epistemic uncertainty-aware adjustments meaningfully mitigate overgeneralization in Mon-MDPs, demonstrating the practicality of our approach.
> ## Questions:
> - **More complex environment**: It is unclear to us what additional scientific value "more complex" environments will add to support our contributions? As stated in Line 238, our goal is not to benchmark Deep RL algorithms, but to study the effect of function approximation when rewards are not always observed outside the tabular setting, where the interplay of partial reward observability, generalization, and overgeneralization emerges clearly. We would appreciate it if the reviewer could help us: what specific insights would more complex environments provide that are not already captured by our current experimental design?
> - **CVaR approximation**: The CVaR cannot be calculated exactly in continuing MDPs and continuing Mon-MDPs with guarantees of convergence, because CVaR depends on the full distribution of returns, not just expectations, and this distribution is extremely complex to represent in sequential decision-making. Consequently, we adopt k-of-N CFR, a principled and tractable approximation with guarantees of convergence to an $\epsilon$-Nash equilibrium when applied to sampled return distributions. This makes it suitable and theoretically justified for our setting.
>
> ## References:
> [1] Chen, K., & Bowling, M. Tractable objectives for robust policy optimization. Advances in Neural Information Processing Systems. 2012

---

> > ### Comment · Reviewer_5gM9 · 2025-11-26
> > **thanks for feedback**
> >
> > I thank the authors for their feedback. I appreciate the clarifications regarding the lack of theory and the environment complexity. On the other hand, I feel that the authors only scratch the surface of the very interesting finding of overgeneralization. The authors argue that there is a fine and thin line between generalization (which can be beneficial and lead to almost-optimal policies) and overgeneralization (which can lead to performance degradation). I feel this is a one of the most interesting findings for practitioners, but it is not covered is significant depth, not only from a theoretical but even from a practical standpoint. For instance, what is the main takeaway message for practitioners who are dealing with such MDPs? Should they just adopt more cautious strategies? What about trying to balance out these two phenomena using cleverer strategies? CVaR is an established framework, and it is certainly a sound mitigation strategy. But in my view the treatment is rather superficial, and I feel this point would deserve a more in-depth analysis. I respect that the authors did not want to focus on formal theory, but then the practical aspects (such as generalisation vs. overgeneralisation), which are of high value to practitioners, should in my view be covered in more depth.

---

> > > ### Author Response · Authors · 2025-11-28
> > > **Authors' Response to Reviewer 5gM9**
> > >
> > > We thank the reviewer for the thoughtful follow-up comments and for acknowledging the clarifications provided in our earlier response.
> > > - **Depth of the overgeneralization findings**: We agree that overgeneralization is one of the most intriguing and practically important phenomena revealed by Mon-MDPs with function approximation. However, overgeneralization is one of the three main contributions in this paper. In Section 4.3, we stated that "overgeneralization is the ratio between the reward model error on the unobservable state-action pairs to the observable ones". This framing makes clear that generalization is a **spectrum**, underfitting at one extreme (ratio $<1$), accurate generalization at the center (ratio $=1$), and overgeneralization at the other extreme (ratio $>1$). One of our goals in this paper is to identify and empirically illustrate this spectrum in Mon-MDPs for the **first time**. A deeper exploration (both theoretical and empirical) would require broader envirionments, algorithmic comparisons, alternative uncertainty-aware methods, and a more analysis of the generalization spectrum broader.
> > > - We revised the paper to discuss the generalization spectrum and to clearly state position this paper as **a first step** in invitigating overgenerlization in Mon-MDPs. We also explicitly note that a more extensive analysis of generalization spectrum is an important direction for future work.
> > > - **Practical guidance for practitioners in Mon-MDPs**: Our results suggest that practitioners can use function approximation in Mon-MDPs to generalize from monitored to unmonitored states while incorporating epistemic-uncertainty–aware methods to mitigate overgeneralization. Beyond adopting “more cautious” strategies, we recommend tha practitioners should actively measure and track epistemic uncertainty to understand when reward models are likely to generalize reliably and when they may fail.
> > > - **CVaR-based mitigation**: In this work, our goal was to demonstrate feasibility rather than to exhaustively analyze mitigation strategies. While CVaR is widely used and theoretically grounded, a more comprehensive comparison with alternative risk-sensitive strategies would indeed strengthen the practical contributions.
> > > - We revised the paper to clearly frame k-of-N CFR as a first step toward a broader treatment of mitigation.

---

### Official Review · Reviewer_ZiWN · 2025-11-05

**Soundness:** 2
**Presentation:** 3
**Contribution:** 2
**Rating:** 4
**Confidence:** 3

**Summary:**

This paper studies function approximation for monitor Mon-MDPs and evaluates its performance across several environments based on the plant-watering setup. The experimental results show that Mon-MDP performs better than several baselines that do not incorporate the Mon-MDP structure with FA. The results also indicate that FA enables better generalization to unseen states when the reward function remains unchanged, but it may lead to over-generalization when the underlying reward function changes. The paper further demonstrates that employing a curiosity-driven learning method can help mitigate over-generalization.

**Strengths:**

- The paper effectively demonstrates the usefulness of function approximation in the context of Mon-MDPs.

- It shows that incorporating robust policy optimization can help handle the over-generalization problem caused by the epistemic uncertainty of the environments.

**Weaknesses:**

- My main concern lies in the novelty and significance of the proposed approach. The algorithm appears to be a combination of several existing techniques, including Mon-MDP, function approximation, and the "learning to be curious" framework to handle out-of-distribution data. While the paper provides extensive experimental results in the plant-watering environment, the findings are not particularly surprising. The advantages of Mon-MDP over baselines have already been shown in the tabular setting. Moreover, the generalization benefits of function approximation are well-known and acknowledged by the authors (line 165). Regarding over-generalization, it is unclear whether this issue is unique to Mon-MDPs or is a general phenomenon associated with FA in standard MDPs. In my understanding, the observed over-generalization likely results from the entirely new patterns not captured in the observed data, a problem typically addressed by robust policy optimization methods. In this sense, it is not evident what new insights or contributions this paper provides to the community.

- The experiments were conducted exclusively in the plant-watering environments. To strengthen the empirical findings, it would be beneficial to include results from more diverse and complex environments.

**Questions:**

- What is the novelty of this work beyond combining Mon-MDP, function approximation, and curiosity-driven learning?

- What unique insights does this study offer about the Mon-MDP problem? (Please refer to the first point of weakness for more details)

---

> ### Author Response · Authors · 2025-11-24
> **Authors' Response to Reviewer ZiWN**
>
> We thank the reviewer for their careful reading and constructive feedback. Below, we address each concern in detail.
> ## Weakness:
> - **Novelty**: While Mon-MDPs and function approximation each have prior foundations, their interaction in the non-tabular Mon-MDP setting introduces challenges that do not arise in any of these components individually. Specifically:
>     - Generalization in Mon-MDPs is fundamentally different from generalization in traditional MDPs. In traditional MDPs, rewards are always observable, making the reward function fully identifiable from experience. In contrast, Mon-MDPs could contain states/actions where the reward is unobservable. As a result, the agent may rely on reward generalization to estimate the environment rewards, then construct the agent's policy using the estimated rewards. This creates behaviors and failure modes (e.g., overgeneralization) that cannot appear in tabular settings or fully observable traditional MDPs.
>     - According to our definition of **Overgeneralization** (line 356), it is tied to Mon-MDPs and reward unobservability. The phenomenon we study is not the typical function approximation overfitting and extrapolation issue. Our definition (line 356) links overgeneralization directly to the agent’s inability to observe rewards. In this sense, overgeneralization is a structural property of Mon-MDPs, not simply a standard function approximation pathology. Tabular Mon-MDPs cannot exhibit this failure mode because they lack function approximation; traditional MDPs cannot exhibit it because rewards are always known.
>     - We agree with the reviewer that the advantage of the reward model over traditional MDP baselines has been shown in the tabular settings, and we mentioned that in the paper (line 287). Although prior work showed advantages of Mon-MDPs in tabular environments, it was not clear whether this advantage survives the complexity of function approximators. Our findings show that reward models remain beneficial under function approximation, but also introduce new pitfalls (e.g., overgeneralizing). This integration between Mon-MDPs and function approximation is novel and was not addressed in prior work.
>     - In summary, our contribution is not the components themselves, but demonstrating and characterizing their interaction in a setting where reward unobservability and generalization are inseparable. This produces insights that cannot be captured in tabular or fully observable settings.
> - **Diverse complex environments**: We appreciate the reviewer’s desire for more diverse benchmarks. However, our goal is to isolate and deeply investigate the core challenge unique to generalization in Mon-MDPs: reward unobservability and its effect on generalization (Line 269). More complex environments introduce confounding factors such as stochastic transitions or partial observability of states (instead of rewards), making it harder to attribute behaviors specifically to reward monitoring. Therefore, by design, the plant-watering environment: i) cleanly separates reward observability,
> ii) exposes generalization failures in a controlled manner, and iii) enables systematic manipulation of monitoring scenarios. We agree that extending to more complex domains is valuable, and we state that as a limitation (line 466).
> ## Questions:
> - The novelty of this paper lies in exploring generalization when rewards are not always observed, and doing so outside the tabular setting. It is unique to study the combined challenges of generalization, overgeneralization, reward observability, and cautious behavior within a single framework (non-tabular Mon-MDPs). While each of these challenges can appear independently in other settings, their interaction emerges naturally in non-tabular Mon-MDPs. This combination provides new insights into how agents learn under partial reward observability.
> - We identify that inference of unobservable rewards (not value estimation) is the primary bottleneck in Mon-MDPs under function approximation. We also show that overgeneralization arises from epistemic uncertainty, making it a Mon-MDP–specific phenomenon rather than a standard function approximation issue. In addition, our experiments hint at the possibility of plasticity loss in Mon-MDPs. Finally, we demonstrate that epistemic uncertainty-aware reward models naturally induce cautious behavior, offering practical insights to mitigate overgeneralization.

---

### Author Response · Authors · 2025-12-03
**Rebuttal Summary**

- We thank all reviewers for their careful evaluations and constructive feedback.
- Across the reviews, there is **broad acknowledgment** that this work:
    - Tackles an **important and previously unexplored problem**: understanding generalization when rewards are not always observable, outside the tabular setting.
    - Identifies **overgeneralization** as a structural phenomenon and showing empirical evidence in terms of epistemic uncertainty and cautious behavior.
    - Provides **targeted, well-designed experiments** that clearly validate core findings while offering the first empirical characterization of overgeneralization in non-tabular Mon-MDPs, with a quantitative definition and analysis.
    - Shows that reward modeling enables learning **near-optimal policies** in Mon-MDPs that are formally unsolvable in the tabular setting.
    - Introduces meaningful insights on how epistemic uncertainty drives reward inference and cautious behavior.
    - Incorporates uncertainty-aware and robust optimization techniques (e.g., risk-averse RL, k-of-N CFR), demonstrating practical mitigation for overgeneralization.
-  Across reviewers, the key concerns relate to **novelty, environment complexity, reward misspecification, and lack of theory**:
    - We clarify that the **novelty** lies not in isolated components but in revealing how generalization, reward observability, and uncertainty interact in a unified framework.
    - **Environment complexity**, we view our simple environments as a **strength** rather than a weakness: they allow us to isolate generalization and reward observability from other confounding factors (e.g., stochastic transitions, partial state observability), while providing controlled conditions to explore generalization and overgeneralization. We explicitly acknowledge that extending our approach to more complex environments is an important future direction.
    - We also address theoretical and mitigation-related concerns by positioning this paper as **a foundational empirical step, necessary** before theory and broader mitigation analyses can be developed.
    - Reward misspecification: We give **a relaxation of the truthful assumption** in this paper; function approximation with an experience replay buffer could potentially cancel out unbiased Gaussian noise. In addition, we added explicit assumptions and discussed noisy/adversarial reward monitoring as an important future work.
- In **conclusion**, this work provides the first empirical investigation of generalization in non-tabular Mon-MDPs, a setting where rewards observability introduces learning challenges fundamentally different from those in traditional MDPs. We identify and formalize overgeneralization as a structural property of Mon-MDPs, distinct from ordinary function-approximation extrapolation, and show how it emerges from the need to infer unobserved rewards. Our experiments reveal new insights into how epistemic uncertainty shapes reward inference, value propagation, and cautious behavior, offering a deeper understanding of when Mon-MDPs succeed or fail under function approximation. Finally, we demonstrate the feasibility of uncertainty-aware mitigation via k-of-N CFR, positioning it as an initial but practically meaningful step toward broader risk-sensitive solutions. Together, these contributions establish a clear and original scientific advance, laying essential groundwork for future theoretical and empirical research on learning under partial reward observability.

---

### Meta-Review · Area_Chair_mXrL · 2026-01-05

**Summary:**

The reviewers found the paper's contribution to be limited. This is partially explained by the paper's presentation, which raised numerous questions. Furthermore, the contribution regarding the overgeneralization phenomenon warrants further investigation. In general, the reviews were fairly constructive and will provide strong support for improving the paper and making its contribution more explicit.

**Reviewer Concerns:**

- `ZiWN` found the contribution limited and mentioned that the empirical evaluation lacked a variety of environments. The rebuttal partially addressed the concerns regarding the limitations of the contribution. While the authors argue that the empirical evaluation is sufficient to gain a deep understanding of the proposed approach, it would be beneficial to demonstrate that the methodology performs well across a variety of environments.
- `5gM9` mentions limited novelty, lack of theory, and simplified problems in the empirical evaluation. The rebuttal partially addressed the concerns of the reviewer.
- `5Zm1`
	- The reviewer had general questions regarding the Mon-MDP model, finding that the experiments were disconnected from the model, and the role of the monitoring reward was unclear. The authors clarified that the experiments consider a sequence of instances of the (general) Mon-MDP and clarified the general role of the monitored reward. Similarly, the monitoring reward played a general role across different monitoring scenarios.
	- The reviewer had questions about the overgeneralization phenomenon and whether it was an issue of distribution shift rather than a problem with the monitoring settings. The rebuttal was insufficient to address the reviewer's concerns, and the paper could study this phenomenon more explicitly.
- `AuFX` was concerned with the lack of theoretical results and the limited scope of the empirical evaluation. Although the rebuttal reasonably argues that this is the paper's position, it did not address these concerns.
- `PqK8`
	- found some of the statements in the paper contradictory. Although the rebuttal clarifies some of these concerns, it is not clear how well they would be addressed in the paper's revision.
	- raised questions regarding the use of an environment with (finite) discrete state and action spaces. The authors argue that the experiments have a large set, which demands function approximation. This partially addresses the reviewer's concern; however, it remains open how the proposed method would handle continuous problems.

**Reviewer Scores:**

- `ZiWN`: 4 -> 4
- `5gM9`: 2 -> 4
- `5Zm1`: 4 -> 4
- `AuFX`: 6 -> 6
- `PqK8`: 2 -> 2

---

### Decision · Program_Chairs · 2026-01-26

Reject